# Analysis of science journalism reveals gender and regional disparities in coverage

Natalie R Davidson, Casey S Greene*

University of Colorado School of Medicine, Aurora, United States

**Abstract** Science journalism is a critical way for the public to learn about and benefit from scientific findings. Such journalism shapes the public's view of the current state of science and legitimizes experts. Journalists can only cite and quote a limited number of sources, who they may discover in their research, including recommendations by other scientists. Biases in either process may influence who is identified and ultimately included as a source. To examine potential biases in science journalism, we analyzed 22,001 non-research articles published by Nature and compared these with Nature-published research articles with respect to predicted gender and name origin. We extracted cited authors' names and those of quoted speakers. While citations and quotations within a piece do not reflect the entire information-gathering process, they can provide insight into the demographics of visible sources. We then predicted gender and name origin of the cited authors and speakers. We compared articles with a comparator set made up of first and last authors within primary research articles in Nature and a subset of Springer Nature articles in the same time period. In our analysis, we found a skew toward quoting men in Nature science journalism. However, quotation is trending toward equal representation at a faster rate than authorship rates in academic publishing. Gender disparity in Nature quotes was dependent on the article type. We found a significant over-representation of names with predicted Celtic/English origin and under-representation of names with a predicted East Asian origin in both in extracted quotes and journal citations but dampened in citations.

*For correspondence:
casey.s.greene@cuanschutz.edu

**Competing interest:** The authors declare that no competing interests exist.

## eLife assessment

This **important** bibliometric analysis shows that authors of scientific papers whose names suggest they are female or East Asian get quoted less often in news stories about their work. While caveats are inevitable in this type of study, the evidence for the authors' claims is **convincing**, with a rigorous, and importantly, reproducible analysis of over 20,000 articles from across 15 years. This paper will be of interest to science journalists and to researchers who study science communication.

## Introduction

Science journalism is an indispensable part of scientific communication and provides an accessible way for everyone from researchers to the public to learn about new scientific findings and to consider their implications. However, it is important to identify the ways in which its coverage may skew toward particular demographics. Coverage of science shapes who is considered a scientist and field expert by both peers and the public. This indication of legitimacy can either help recognize people who are typically overlooked due to systemic biases or intensify biases. Journalistic biases in general-interest, online and printed news have been observed by journalists themselves (*French, 2016*; *LaFrance, 2013*; *LaFrance, 2016*; *Yong, 2018*), as well as by independent researchers (*Shor et al., 2015*; *Shor*

*et al., 2014*; *Ross and Carter, 2011*; *Jia et al., 2016*; *Layton and Shepard, 2013*; *Who Makes the News, 2024*; *Holman et al., 2018*). Researchers found a gap between men and women subjects or sources, with independent studies finding that between 17% and 40% of total subjects were women across multiple general-interest printed news outlets between 1985 and 2015 (*Shor et al., 2015*; *Shor et al., 2014*; *Who Makes the News, 2024*). One study found 27–35% of total subjects in international science and health-related news were women between 1995 and 2015, and 46% in print, radio, and television in the United States in 2015 (*Who Makes the News, 2024*). While gender disparities in news coverage have been extensively researched, our research is different because it focuses on science journalism and comparing it against the demographics of actively publishing scientists. Additionally, our work focuses on research into disparities with respect to name origins, a focus which is currently lacking in the literature.

It should be noted that scientific news coverage is confounded by the existing differences in gender and racial demographics within the scientific field (*NSF - National Science Foundation, 2023*; *Iwasaki, 2019*). However, we are interested in quantifying disparities with respect to observed demographic differences in the scientific field, using academic authorship as an estimate for the existing demographics. This is similar to other studies that have quantified gender or racial disparities in science as observed in citation (*King et al., 2017*; *Larivière et al., 2013*) and funding rates (*Stevens et al., 2021*; *Erosheva et al., 2020*; *Hoppe et al., 2019*; *Ley and Hamilton, 2008*; *Ginther et al., 2011*).

In researching a story, a journalist will typically interview multiple sources for their opinion, potentially asking for additional sources, thus allowing individual unconscious biases at any point along the interview chain to skew scientific coverage broadly. In addition, the repeated selection of a small set of field experts or the approach a journalist takes in establishing a new source may intensify existing biases (*LaFrance, 2016*; *Yong, 2018*; *Selby, 2016*). While disparities in representation may go unnoticed in a single article, analyzing a large corpus of articles can identify and quantify these disparities and help guide institutional and individual self-reflection. In the same vein as previous media studies (*Shor et al., 2015*; *Shor et al., 2014*; *Ross and Carter, 2011*; *Jia et al., 2016*; *Layton and Shepard, 2013*; *Who Makes the News, 2024*), we sought to quantify differences in representation across predicted of gender and name origin beyond the existing demographic differences in the scientific field. Our study focused solely on science journalism, specifically content published by *Nature*. Since *Nature* also publishes primary research articles, we used these data to determine the demographics of the expected set of possible sources. This is not a perfect comparator since journalists will not cover every research article presented in the journal. However, we assume it is a reasonable baseline and that large deviations are worthy of investigation as reflecting potential journalistic biases. For clarity, throughout this manuscript we will refer to journalistic articles as *news* and academic, primary research articles as *papers*. Furthermore, when we refer to 'authors' we mean authors of academic papers, not journalists. In our analysis, we identified quoted and cited people by analyzing the content and citations within all news articles from 2005 to 2020, and compared this demographic to the academic publishing demographic by analyzing first and last authorship statistics across all *Nature* papers during the same time period.

Through our analysis of 22,001 news articles, we were able to identify >88,000 quotes and >15,000 citations with sufficient speaker or author information. We also identified first and last authors of >10,000 *Nature* papers. We then identified possible differences in predicted gender or name origin using the extracted names. The extracted names were used to generate three data types: quoted, mentioned, and cited people. We used computational methods to predict gender and identified a trend toward quotes from people predicted to be men in news articles when compared to both the general population and authors predicted to be men in papers. Within the period that we examined, the proportion of quotes predicted to be attributed to men in news articles went from initially higher to currently lower than the proportion first and last authors predicted to be men in *Nature* papers. Furthermore, we found that the quote difference was dependent on article type; the 'Career Feature' column achieved gender parity in quoted speakers.

We also used computational methods to predict name origins of quoted, mentioned, and cited people. Through our analysis, we found a significant over-representation of names with predicted Celtic/English origin and under-representation of names with a predicted East Asian origin in both quotes and mentions. To our knowledge, our work is the first to identify a substantial under-representation of names with a predicted East Asian origin in scientific journalism.

While we focused on news from *Nature*, our software can be repurposed to analyze other text. We hope that publishers will welcome systems to identify disparities and use them to improve representation in journalism. Furthermore, our approach is limited by the features we were able to extract, which only reflects a portion of the journalistic process. Journalists could additionally track all sources they contact to self-audit. However, auditing is only part of the solution; journalists and source recommenders must also change their source gathering patterns. To help change these patterns, there exist guides (*Selby, 2016*), databases (*Women's Media Center, 2024*), and affinity groups that can help us all expand our vision of who can be considered a field expert.

## Results
### Creation of an annotated news dataset

We have analyzed the text of 22,001 news-related articles hosted on 'https://www.nature.com/' that span 15 years from 2005 to 2020. Our primary focus is on 16,080 articles written by journalists which include the following five article types: 'Career Feature', 'News', 'News Feature', 'Technology Feature', and 'Toolbox'. 'Career Feature' generally focuses on the career-related aspects of being a scientist. 'News' and 'News Feature' focuses on current events related to science as well as new scientific findings. It should be noted that the types of articles contained in 'News' changed over time

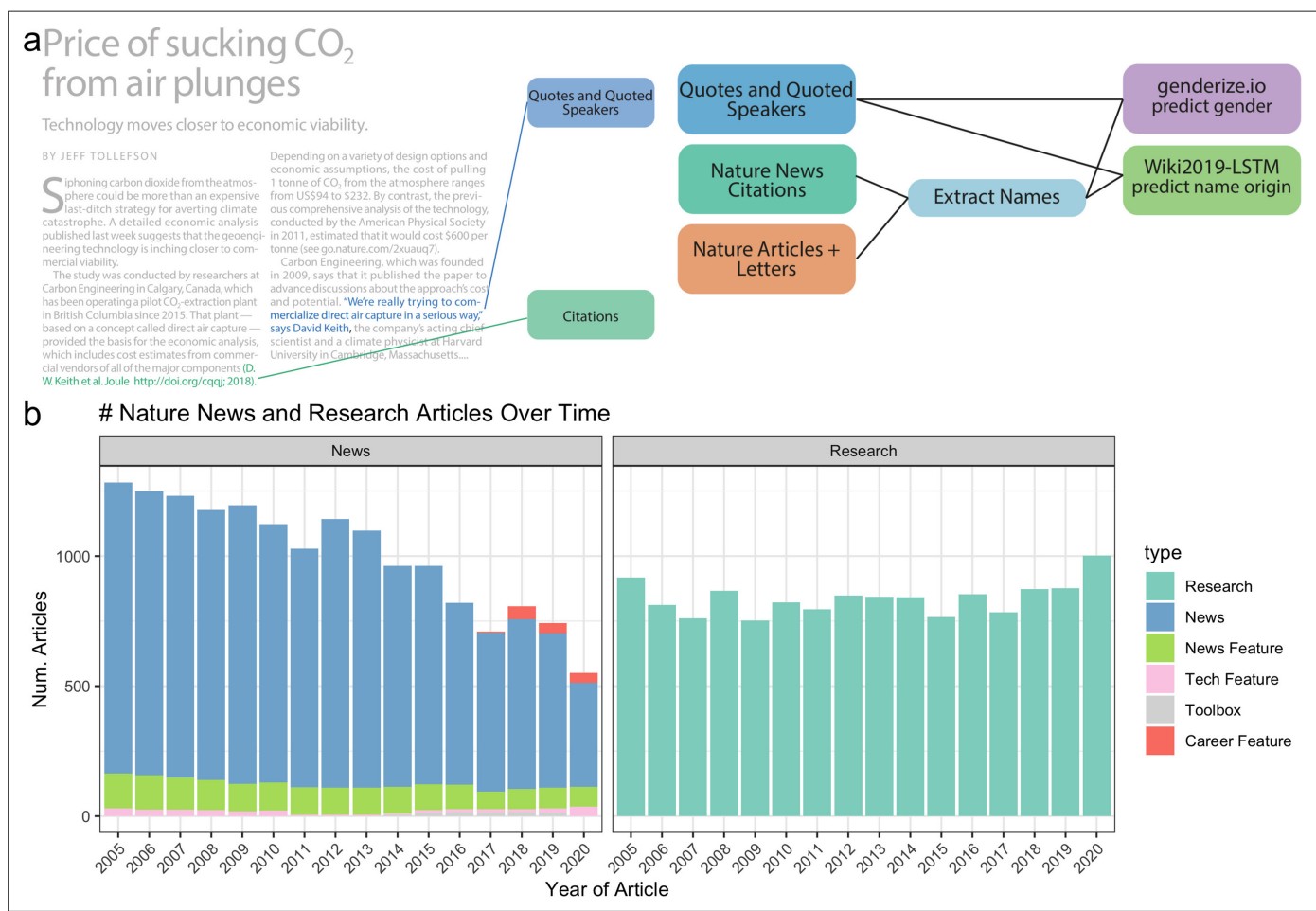

**Figure 1.** Data and processing pipeline overview. (**a**), left, depicts an example news article and the type of data extracted from the text. Green and blue highlighted text depicts all quotes, and associated speakers identified by the coreNLP pipeline. A custom script described in section *Methods* identifies all citations. (**a**), right, charts the analyses done on the extracted names and locations from news articles and papers published by Nature. (**b**) shows the types and amounts of articles that we have used for analyses.

The online version of this article includes the following figure supplement(s) for figure 1:

**Figure supplement 1.** Benchmark data.

**Table 1.** Breakdown of quotes at major processing steps.

| Processing step | Frequency |
|---|---|
| Total quotes | 105,457 |
| Quotes with a full name or pronoun associated | 96,620 |
| Quotes with a gender prediction | 96,390 |
| Quote with a full name | 88,535 |
| Quotes with a name origin prediction | 100,457 |

which may induce content shifts in a subset of the articles within our corpus. 'Technology Feature' also covers current events and scientific findings, but additionally focuses on how science intersects with technology, such as apps, methodologies, tools, and practices. Lastly, 'Toolbox' is similar to 'Technology Feature', but is more centered on technology, especially the tools used to perform science. We also include one analysis of the scientist-written news articles, 'Career Column' and 'News and Views', as an additional set of 5921 articles. 'Career Column' is similar to 'Career Feature', except it is not written by journalists, but individuals in the scientific field. 'News and Views' is similar to a review article, where a field expert writes an article relating to a recently written article within Nature.

The top 3 observed article frequencies are 'Research' (including 'Letters' and 'Articles'), 'News', and 'News Feature'. Since *Nature* merged 'Letters' and 'Research' papers in 2019, we combined them in our analysis. We observed substantial variability in the number of *Nature* news articles by type between 2005 and 2020 (*Figure 1b*). The changing classification of article types partially explains temporal changes in the frequency of news articles within each category. Over time, the frequency of 'News' articles decreased; however, more specific news-related article types increased, including the introduction of the new categories 'Career Feature', 'Toolbox', and 'Career Column.

## Extracted data used for analysis

The text and citations were then uniformly processed as depicted in *Figure 1a* to identify: (1) quotes and quoted speakers (blue box) and (2) cited authors (green box). The extracted names from the text were used to generate three data types for downstream processing: quoted, mentioned, and cited people. Quoted names are any names that were attached to a quote within the article. Mentioned names are any names that were stated within the article. Cited names are all author names of a scientific paper that was cited in the news article. A summary of frequencies for each data type at each point of processing is provided in *Tables 1–4*. We scraped the text using the web-crawling framework Scrapy (*Scrapy developers, 2020*), processed, and ran it through the coreNLP pipeline (Methods). To identify quotes and speakers, we used the coreNLP quote extraction and attribution annotator. We performed multiple name formatting processes (Methods) to identify the speaker's full name for gender and name origin prediction. All names where we could identify two name parts, assumed to be a first and last names exclusive of titles, were used for gender prediction and checked against the genderize.io database. Since names used in the name origin analysis were computationally analyzed and not checked for existence in an existing database, we used additional filters (Methods). All names excluded from the name origin analysis of quotes and mentions are provided on our github in the file 'data/author_data/all_mentioned_fullname_excluded.tsv' and 'data/author_data/all_speaker_fullname_excluded.tsv'. We found that most names were excluded because two name parts, assumed to be a first and last names exclusive of titles, were not found. We scraped the citations using an

**Table 2.** Breakdown of citations at major processing steps.

| Writer of article | Total citations | Total Springer Nature citations | First author citations with a full name | Last author citations with a full name | First author citations with a name origin prediciton | Last author citations with a name origin prediciton |
|---|---|---|---|---|---|---|
| Journalist | 15,713 | 5736 | 4452 | 4464 | 4449 | 4447 |
| Scientist | 40,707 | 14,597 | 11,276 | 11,170 | 11,276 | 11,152 |

**Table 3.** Breakdown of all Springer Nature papers at major processing steps.

| Processing step | Frequency |
| --- | --- |
| # Springer Nature articles | 38,400 |
| # First + last authors with a full name in Springer Nature articles | 55,370 |
| # First + last authors with a gender prediction in Springer Nature articles | 51,686 |
| # First + last authors with a name origin prediction in Springer Nature articles | 55,197 |

independent scraper to the text scraper, but still utilizing the Scrapy framework. All identified DOI's were queried using the *Springer Nature* API to attain all authors' names, positions, and affiliations.

## Comparator datasets

Next, we determined if the quoted speakers and cited authors in news articles have a similar demographic makeup as the scientists who publish their primary research in *Nature*. To make this determination, we used all authors' names, positions, and affiliations of papers published by *Nature* over the same time period (*Figure 1a*, dark orange box). The author metadata of *Nature* papers from 2005 to 2020 totaled 13,414. To more broadly represent overall science authorship, we also separately analyzed 38,400 *Springer Nature*-published papers from English-language journals over the same time. It should be noted that extracted quotes may come from multiple types of people, such as academic scientists, clinicians, the broader scientific community, politicians, and more. However, through anecdotal observation we believe that most sources come from either academic scientists or those actively involved in science. Similarly, author names were uniformly processed and then used to predict both gender and name origin.

## Quoted speakers and primary research authors in *Nature* are more often men

To quantify and compare the gender demographic of quoted people and authors, we analyzed their names. While we could have analyzed the proportion of unique predicted men speakers, we were interested in measuring the overall participation rates by gender and analyzed the proportion of total quotes, e.g. a single speaker may have more than one quote in an article. Furthermore, we assume that a majority of quoted speakers are typically involved in scientific research and therefore primary research authors is a comparable demographic. *Figure 2* shows an overview of the process and example input data for this analysis: (1) quotes and quoted speakers (blue box) and (2) first and last authors' names of papers published by *Nature* (dark orange box). These analyses relied upon accurate gender prediction of both authors and speakers. To predict the gender of the speaker or author, we used the package genderizeR (*Wais, 2016*), an R package wrapper to access the genderize.io API (*Demografix ApS, 2024*) to get binary gender predictions for each identified first name. We unfortunately cannot identify non-binary gender expression with the tools we used. Performance of binary prediction was evaluated on a benchmark dataset of 30 randomly selected news articles, 10 from each of the following years: 2005, 2010, and 2015 (*Figure 1—figure supplement 1*). In addition, genderize.io has been found by independent researchers to have an error rate comparable to other published gender prediction methods, with a error rate on predicted names below 6% (*Santamaría and Mihaljević, 2018*; *Sebo, 2021*). However, it should be noted that the error rate varies by name origin with the largest decrease in performance on names with an Asian origin (*Santamaría and Mihaljević,*

**Table 4.** Breakdown of all Nature papers at major processing steps.

| Processing step | Frequency |
| --- | --- |
| # Nature articles | 13,414 |
| # First + last authors with a full name in Nature articles | 21,996 |
| # First + last authors with a gender prediction in Nature articles | 21,173 |
| # First + last authors with a name origin prediction in Nature articles | 21,996 |

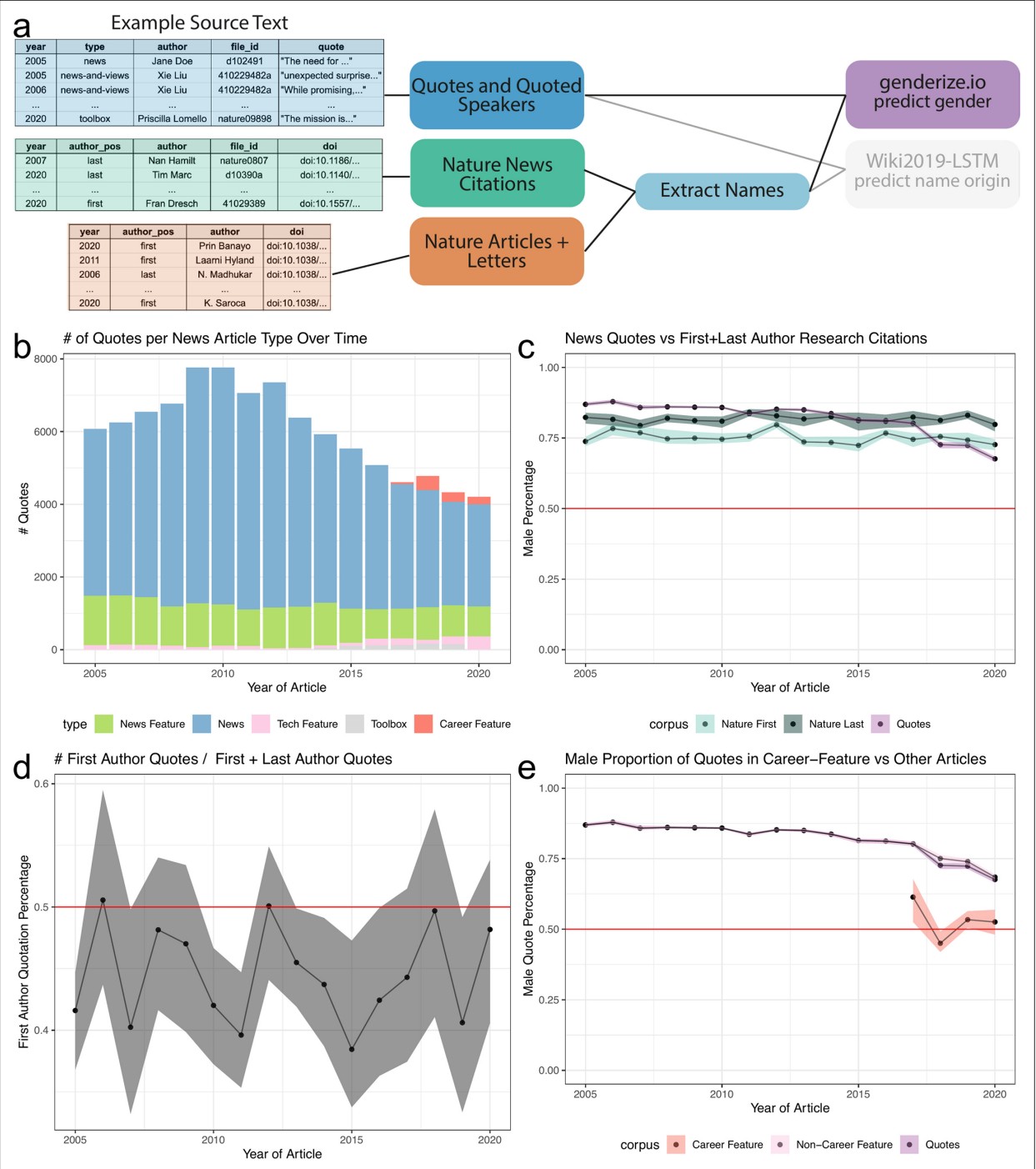

**Figure 2.** Speakers predicted to be men are sometimes over-represented in quotes, but this depends on the year and article type. (**a**), left, depicts an example of the names extracted from quoted speakers in news articles and authors in papers. (**a**), right, highlighted the data types and processes used to analyze the predicted gender of extracted names. (**b**) shows an overview of the number of quotes extracted for each article type. (**c**) depicts three trend lines: purple: proportion of quotes for a speaker estimated to be a man; light blue: proportion of first author papers estimated to be a man; dark blue: proportion of last authors predicted to be a man. We observe that the proportion of quotes estimated to come from a man is steadily decreasing, most notably from 2017 onward. This decreasing trend is not due to a change in quotes from the first or last author, as observed in (**d**). (**d**) shows a consistent but slight bias toward quoting the last author of a cited article than the first author over time. (**e**) depicts the frequency of quote by article type highlighting an increase in quotes from 'Career Feature' articles. (**e**) depicts that the quotes obtained in this article type have reached parity. The colored bands represent a 5th and 95th bootstrap quantiles in all plots, and the point is the mean calculated from 1000 bootstrap samples.

The online version of this article includes the following figure supplement(s) for figure 2:

*Figure 2 continued on next page*

**Figure supplement 1.** Speakers predicted to be men are over-represented in news quotes regardless of predicted journalist gender.

**Figure supplement 2.** Speakers predicted to be men are over-represented in news quotes when compared against Springer Nature authorship.

*2018*; *Sebo, 2021*). In our analysis, we did not observe a large difference in names predicted to come from a man or woman between predicted East Asian and other name origins (*Table 5*).

We first examined the number of quotes identified within each type of science news article (*Figure 2b*), totaling 105,457 quotes with 96,390 of them containing a gender prediction for the speaker. Quote frequencies vary by article type. We compared the number of quotes from predicted men to the number of predicted first and last author men published in *Nature*. The total number of authors with a gender prediction was 10,601 first authors and 10,572 last authors across a total of 11,161 publications. As denoted by the red line, we found that the predicted genders of authors and source quotes were far from gender parity (*Figure 2c*). We found this result consistent for articles written by either a predicted man or woman journalist (*Figure 2—figure supplement 1a and b*). Additionally, we observed a difference in the predicted genders between first and last authors, with the last authors more frequently predicted to be a man. In our supplemental analyses, we provide an additional comparator, a selection of articles from English-language journals published by *Springer Nature* (*Figure 2—figure supplement 2a*). The predicted gender gap between first and last authors was larger in our selection of *Springer Nature* papers; however, both first and last authors were predicted to be closer to parity than for *Nature* authors. Overall, predicted men were more frequently quoted than predicted women in *Nature* news articles and first and last authors in *Nature* and *Springer Nature* papers over the same time period.

## Career feature articles reach gender parity

The gender proportions of authorship were relatively stable over time for both *Nature* and *Springer Nature* papers. In contrast, we found that the rate of quotes predicted to be from men noticeably decreased over time. In 2005, the fraction of quotes predicted to be from men was 87.09% (5291/6075), whereas in 2020 it was 68.86% (2870/4168). We identified that a large decrease occurred in *Nature* between 2017 and 2018. We explored the possible reasons for this decrease. First, we looked at the authorship position of speakers who were quoted about their published paper (*Figure 2d*). We identified 6545 quotes with an associated citation (2871 first author and 3674 last author quotes). We found that quotes are slightly biased toward last authors from 2005 to 2020, but because the fraction of last authors predicted to be men remained stable over time both for *Nature* and the selection of *Springer Nature* papers, which likely does not explain the downward trend. We then analyzed the breakdown of gender in quotes by article type. Interestingly, one article type, 'Career Feature', achieved gender parity in its quotes (*Figure 2e* and *Figure 2—figure supplement 2b*). In this article type, we identified a total of 898 quotes (449 predicted women's and 449 predicted men's quotes), which only slightly pulled the overall quote gender ratio closer to parity from 2018 onward. In general, we found that each article type independently trended toward gender parity.

## Predicted Celtic English name origins are over-represented in quoted and mentioned people, while predicted East Asian name origins are under-represented

To identify possible disparities with respect to name origin, we again used the extracted names of quoted speakers from *Nature* news articles and last authors of published papers in *Nature*. In addition, we also identified the last authors of all papers cited by a *Nature* news article. All

**Table 5.** Quoted speaker gender by name origin.

|  | Women | Men | Proportion men |
|---|---|---|---|
| African | 270 | 1554 | 0.8519737 |
| ArabTurkPers | 346 | 1765 | 0.8360966 |
| CelticEnglish | 6399 | 33,329 | 0.8389297 |
| EastAsian | 1090 | 4438 | 0.8028220 |
| European | 4788 | 22,844 | 0.8267226 |
| Greek | 73 | 445 | 0.8590734 |
| Hebrew | 213 | 1303 | 0.8594987 |
| Hispanic | 760 | 2450 | 0.7632399 |
| Nordic | 593 | 2397 | 0.8016722 |
| SouthAsian | 465 | 2019 | 0.8128019 |

processed names were then input into Wiki2019-LSTM and assigned one of 10 possible name origins (Methods). In our analysis, we use name origin to estimate the perceived ethnicity of a primary source by a journalist or fellow scientists who might recommend the individual as a source. Our prediction is not intended to assign ethnicity to an individual, but to be used broadly as a tool to quantify representational differences in a journalist's sociologically constructed perception of a primary source's ethnicity. *Figure 3a* shows an overview of the process and example input data for this analysis: (1) quotes and quoted speakers (blue box), (2) names of cited first and last authors in news articles (green), and (3) first and last authors' names of papers published by *Nature* (dark orange box). We divided our analysis into three parts: firstly, quantifying the proportions of predicted name origins of first and last authors cited in *Nature* news articles. Secondly, calculating the proportion of quotes from speakers with a predicted name origin. Thirdly, calculating the proportion of unique names mentioned within an article with a predicted name origin. As a comparator set, we again used the first and last author names in *Nature* papers for all three analyses. Additionally, in our supplemental analyses, we compared against the first and last authorships in a selection of *Springer Nature* papers. We found that the number of quotes and unique names mentioned dramatically outnumbered the number of cited authors in *Nature* news articles, as well as first and last authors within *Nature* papers (*Figure 3— figure supplement 1a*). Still, since we have more than 100 observations per time point for each data type, we believe this is sufficient for our analysis. Minimum and median per data type over all years: *Nature* papers (568, 684); *Springer Nature* papers (1332, 1710); *Nature* quotes (3788, 5696); *Nature* mentions (3225, 4752); citations in journalist-written *Nature* article (142, 268) citations in a scientist-written *Nature* article (512, 664).

## News citation rates across name origin predictions nearly match *Nature* authorship

In comparing the citation rate of first and last author name origins in news articles, we decided to additionally analyze scientist-written articles. Though fewer in number, scientist-written news articles have many citations, making the set sufficient for analysis and providing an opportunity to measure differences in citation patterns between journalists and scientists. In both journalist- and scientist-written articles, we found that most cited name origins were predicted Celtic/English or European, both with a bootstrapped estimated citation rate between 17.9% and 39.6% (*Figure 3—figure supplement 1b, c*). East Asian predicted name origins are the third highest proportion of cited names, with a bootstrapped estimated citation rate between 7.3% and 28.1%. All other predicted name origins individually account for less than 8.1% of total cited authors.

We analyzed how these distributions compare to the composition of the first and last authors in *Nature* (*Figure 3—figure supplement 2*), by examining the top three most frequent predicted name origins (*Figure 3b, c*, *Table 6*). When considering only first authors, we found a slight over-representation for predicted Celtic/English name origins and a small under-representation for predicted East Asian name origins in scientist- and journalist-written news articles when compared to the composition of first authors in *Nature* (*Figure 3b, c*). When considering last authors, this pattern no longer exists. Furthermore, we found no substantial difference for European or other predicted name origins when comparing against first and last authorships within *Nature* (*Figure 3—figure*

**Table 6.** Mean fold change comparison with Nature from bootstrap samples with 95% CI.

|  | CelticEnglish | EastAsian | European |
|---|---|---|---|
| citation_journalist_first vs. nature_first | 1.36 (0.96, 1.74) | 0.7 (0.46, 0.91) | 1.01 (0.8, 1.25) |
| citation_journalist_last vs. nature_last | 1.18 (0.93, 1.54) | 0.82 (0.42, 1.27) | 0.93 (0.71, 1.19) |
| citation_scientist_first vs. nature_first | 1.26 (1.05, 1.5) | 0.81 (0.66, 1.02) | 1.05 (0.88, 1.22) |
| citation_scientist_last vs. nature_last | 1.11 (0.95, 1.31) | 0.77 (0.58, 0.99) | 1.06 (0.93, 1.19) |
| quote vs. nature_first | 2.12 (1.77, 2.51) | 0.25 (0.2, 0.32) | 1.01 (0.81, 1.22) |
| quote vs. nature_last | 1.52 (1.32 1.75) | 0.39 (0.3, 0.49) | 0.89 (0.79, 1.01) |
| mention vs. nature_first | 2.03 (1.67, 2.39) | 0.29 (0.23, 0.36) | 1.02 (0.81, 1.22) |
| mention vs. nature_last | 1.44 (1.26, 1.67) | 0.45 (0.35, 0.54) | 0.89 (0.79, 1) |

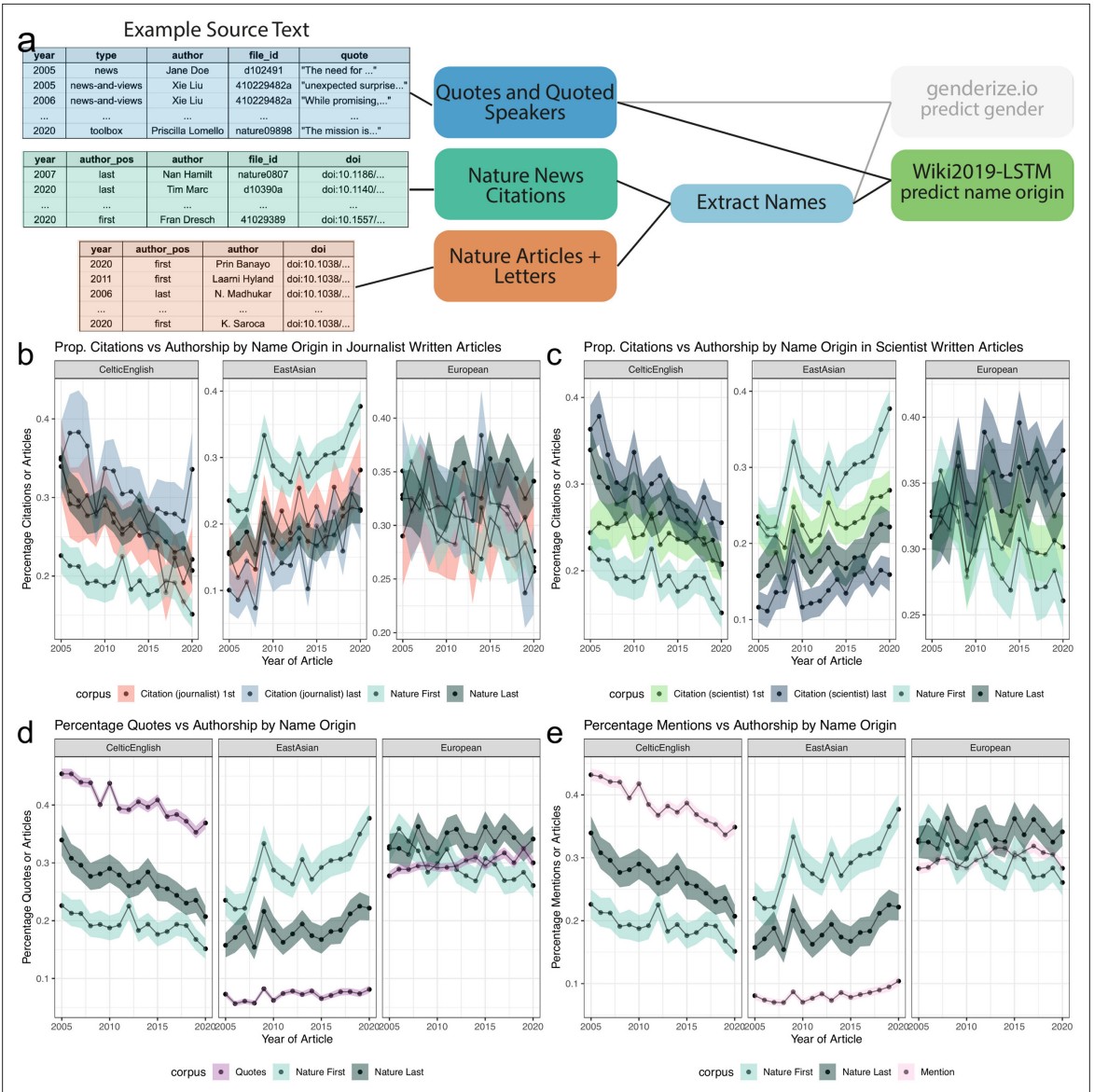

**Figure 3.** Analysis of quotes and citations found over-representation of Celtic/English and under-representation of East Asian predicted name origins. (**a**), left, depicts an example of the names extracted from quoted speakers and citations found within news articles and authors in papers. (**a**), right, highlights the data types and processes used to analyze the predicted origin of extracted names. (**b and c**) depict a comparison between the predicted name origins of last authors in Nature and cited papers in the news. (**b and c**) differ in the news article types. (**b**) calculates the predicted name origin proportion using only journalist-written articles, whereas (**c**) only uses scientist-written articles. The distinction between scientist- and journalist-written articles are defined by the article appearing in either the 'Career Column' or 'News and Views' sections, or another section, respectively. Similarly, (Panels **d and e**) depict two possible trend lines, comparing predicted name origins of either quoted or mentioned people against name origins of last authors of Nature research papers. For more precise numerical comparisons, the mean yearly fold change for each comparison is provided in *Table 6*.

The online version of this article includes the following figure supplement(s) for figure 3:

**Figure supplement 1.** Predicted Celtic/English, and European name origins are the highest cited, quoted, and mentioned.

**Figure supplement 2.** Distribution of name origins Nature and Springer Nature articles.

**Figure supplement 3.** Over-representation of predicted Celtic/English and under-representation of East Asian name origins are also found in comparison to Nature and Springer Nature articles.

**Figure supplement 4.** Over-representation of predicted Celtic/English and under-representation of East Asian quotes and mentions are reduced when additionally considering citation (**a–d**) depicts twelve plots, each for a possible name origin comparison against a background set.

**Table 7.** Mean fold change comparison with Springer Nature from bootstrap samples with 95% CI.

| | CelticEnglish | EastAsian | European |
|---|---|---|---|
| citation_journalist_first vs. springer_first | 1.99 (1.42, 2.64) | 0.69 (0.47, 0.96) | 1.14 (0.89, 1.47) |
| citation_journalist_last vs. springer_last | 2.01 (1.31, 3.08) | 0.56 (0.3, 0.82) | 1.12 (0.91, 1.37) |
| citation_scientist_first vs. springer_last | 1.54 (0.95, 2.17) | 0.91 (0.62, 1.64) | 1.13 (0.91, 1.93) |
| citation_scientist_last vs. nature_last | 1.11 (0.95, 1.31) | 0.77 (0.58, 0.99) | 1.06 (0.93, 1.19) |
| quote vs. springer_last | 2.58 (1.74, 3.6) | 0.28 (0.2, 0.54) | 1.08 (0.84, 1.35) |
| quote vs. nature_last | 1.52 (1.32, 1.75) | 0.39 (0.3, 0.49) | 0.89 (0.79, 1.0) |
| mention vs. springer_last | 2.45 (1.65, 3.42) | 0.32 (0.23, 0.59) | 1.08 (0.85, 1.32) |
| mention vs. nature_last | 1.44 (1.26, 1.67) | 0.45 (0.35, 0.54) | 0.89 (0.79, 1) |

*supplement 3a*, *Table 6*). We also observed the predicted Celtic/English over-representation and East Asian under-representation when considering our subset of *Springer Nature* papers (*Figure 3— figure supplement 3b*, *Table 7*) for both journalist- and scientist-written news articles. In contrast to *Nature*, in the *Springer Nature* set, we see a difference in predicted European name origins, with a growing over-representation. Additionally, we see a difference in predicted Arabic/Turkish/Persian name origins frequencies between cited authors and *Springer Nature* authors, however the absolute difference is lower than observed for Celtic/English and East Asian predicted name origins.

## News quotation rates are over-represented for predicted Celtic English and under-represented for East Asian name origins

We then sought to determine whether or not the quoted speaker demographic replicated the cited authors' over- and under-representation patterns. We found a much stronger Celtic/English over-representation in comparison to citation patterns, with quotes from those with Celtic/English name origins at a much higher frequency than quotes from those with European name origins (*Figure 3— figure supplement 1d*, *Table 6*). We also found a much stronger reduction of quotes from people with predicted East Asian name origins (*Figure 3—figure supplement 1b*), with never more than 8.2% of quotes (*Figure 3d*, *Table 6*). This reveals a large disparity when considering that people with a predicted East Asian name origin constitute between 7.3% and 24.6% of last authors cited in either journalist- or scientist-written news articles (*Figure 3b and c*, *Table 6*). When we compare against first and last authorships in *Nature* across all predicted name origins, we find that for all other name origins except for East Asian and Celtic/English, the quote rates closely match the predicted name origin rate of first and last authors in *Nature* (*Figure 3—figure supplement 3c*, dark gray and light blue lines compare to the purple lines).

To further understand the source of Celtic/English over-representation and East Asian under-representation, we selected a subset of quotes from people whose works were also cited in the news article. The purpose of this additional comparison of quoted speakers versus quoted *and* cited speakers was to reveal source gathering patterns beyond cited works. We found that the proportion of predicted East Asian name origins was closer to the expected rate after considering only quoted speakers with citations, more closely matching the analysis on citations alone (*Figure 3—figure supplement 4a and b*). This indicates that expert opinions gathered beyond manuscript authors are responsible for a large proportion of the observed name disparities.

Next, we sought to determine if predicted journalist name origin had any effect on quote disparities. We found that journalists with a predicted East Asian name origin had a higher rate of East Asian quoted speakers (24.3%) in comparison to journalists with Celtic/English (3.8%) or European (8.6%) predicted name origins (*Table 8*). To examine if this was again driven by source gathering beyond manuscript authors we again subsetted the quotes by adding two constraints: (1) the quotes must be from a cited first or last author in the same news article (*Table 9*) and (2) that the cited article must have a US affiliation (*Table 10*). We found that differences between journalists with different predicted name origins was nearly eliminated when restricting to quoted and cited speakers, and with the additional restriction of US-affiliated citations, as evidenced in the predicted East Asian column

**Table 8.** Quoted speaker name origin, by journalist name origin.

| Journalist name origin | African | Arab Turk Pers | Celtic English | East Asian | European | Greek | Hebrew | Hispanic | Nordic | South Asian |
|---|---|---|---|---|---|---|---|---|---|---|
| CelticEnglish | 0.020 | 0.025 | 0.484 | 0.038 | 0.319 | 0.006 | 0.016 | 0.033 | 0.035 | 0.022 |
| EastAsian | 0.018 | 0.017 | 0.354 | 0.243 | 0.250 | 0.004 | 0.016 | 0.026 | 0.036 | 0.035 |
| European | 0.022 | 0.023 | 0.420 | 0.086 | 0.326 | 0.005 | 0.016 | 0.043 | 0.032 | 0.027 |

of *Table 10*. The differences between *Table 8* and *Tables 9 and 10* indicate that the predicted name origin of a journalist has some association with sources gathered outside of directly cited works.

When comparing *Nature* articles against the *Springer Nature* set of first or last author, we again find the same patterns in quoted speakers with East Asian, Celtic/English, and Arabic/Turkish/Persian predicted name origins as we did in the previous citation analysis (*Figure 3—figure supplement 3d*, green and purple lines). In addition, we find an under-representation of predicted Hispanic, South Asian, and Hebrew name origins when comparing against the predicted name origin rate of first and last authors in our *Springer Nature* set.

## News mention rates are over-represented for predicted Celtic English and under-represented for East Asian name origins

Since many journalists use additional sources that are not directly quoted, we also analyzed likely paraphrased speakers, e.g. a case in which the person was a source and mentioned in the story but not directly quoted. To do this, we identified all unique names that appeared in an article, which we term *mentions*. We found the same pattern of over-representation for predicted Celtic/English name origins and under-representation for East Asian name origins when comparing against both *Nature* and *Springer Nature* first and last authorships (*Figure 3e*, *Figure 3—figure supplement 1d, e*, *Figure 3—figure supplement 3e, f*, *Table 6*, *Table 7*). Similar to the quote analysis, we selected a subset of mentions from people that were also cited in the news article. We again found that the disparity was greatly reduced (*Figure 3—figure supplement 4c, d*).

## Discussion

Science journalism is the critical conduit between the academic and public spheres and consequently shapes the public's view of science and scientists. However, as observed in other forms of recognition in science, biases may shift coverage away from the known demographics within science (*Jan, 2022*). Ideally, scientific journalism is representative of academic papers. Though it would be best for news coverage to promote equitable representation, at a minimum, quotes and citations would ideally match the predicted name origin and gender demographics of scientific academia. To examine this last point, we analyzed 22,001 news articles published in *Nature*, to identify quoted, mentioned, and cited people. We then compared this to the authorship statistics from *Nature*'s papers and a subset of *Springer Nature*'s English-language papers.

We first looked at possible gender differences in quotes and found a large, but decreasing, gender gap when compared to the general population in all but one article type. Additionally, this result was consistent in articles written by journalists predicted to be women or men. We found that one column, 'Career Feature', has an equal number of quotes from both genders, showing that gender parity is possible in science journalism. This finding, coupled with the near equal number of articles written by journalists predicted to be men or women, argues for more diversity in topical coverage.

**Table 9.** Quoted + cited speaker name origin, by journalist name origin.

| Journalist name origin | African | Arab Turk Pers | Celtic English | East Asian | European | Greek | Hebrew | Hispanic | Nordic | South Asian |
|---|---|---|---|---|---|---|---|---|---|---|
| CelticEnglish | 0.016 | 0.027 | 0.368 | 0.070 | 0.363 | 0.008 | 0.017 | 0.023 | 0.083 | 0.025 |
| EastAsian | 0.002 | 0.077 | 0.377 | 0.143 | 0.167 | 0.000 | 0.012 | 0.133 | 0.019 | 0.080 |
| European | 0.014 | 0.028 | 0.363 | 0.116 | 0.352 | 0.006 | 0.030 | 0.026 | 0.035 | 0.030 |

**Table 10.** Quoted speakers (with US-affiliated citation) name origin, by journalist name origin.

| Journalist name origin | African | Arab Turk Pers | Celtic English | East Asian | European | Greek | Hebrew | Hispanic | Nordic | South Asian |
|---|---|---|---|---|---|---|---|---|---|---|
| CelticEnglish | 0.011 | 0.023 | 0.378 | 0.086 | 0.361 | 0.010 | 0.021 | 0.029 | 0.056 | 0.025 |
| EastAsian | 0.000 | 0.066 | 0.340 | 0.148 | 0.209 | 0.000 | 0.005 | 0.148 | 0.033 | 0.049 |
| European | 0.021 | 0.030 | 0.410 | 0.111 | 0.300 | 0.012 | 0.023 | 0.019 | 0.030 | 0.046 |

'Career Feature' articles highlight current topics relevant to working scientists and frequently highlight systemic issues with the scientific environment. This column allows space for marginalized people to critique the current state of affairs in science or share their personal stories. This type of content encourages the journalist to seek out a diverse set of primary sources. Including more content that is not primarily focused on recent publications, but all topics surrounding the practice of science, can serve as an additional tool to rapidly achieve gender parity in journalistic recognition. However, we note that not all forms of recognition are the same; researchers may prefer to have their work featured instead of personal stories or critiques of the scientific environment.

When considering the relative proportion of authors and speakers predicted to be men, we only find a slight over-representation of men. This over-representation is dependent on the authorship position and the year. Before 2010, quotes predicted as from men are over-represented in comparison to both first and last authors, but between 2010 and 2017 quotes predicted from men are only over-represented in comparison for first authors. In 2020, we find a slight over-representation of quotes predicted to be from women relative to first and last authors, but still severely under-represented when considering the general population. The choice of comparison between first and last authors can reveal different aspects of the current state of academia. While this does not hold in all scientific fields, first authors are typically early career scientists and last authors are more senior scientists. It has also been shown that early career scientists tend to be more diverse than senior scientists (*Bankston et al., 2020*; *Nikaj et al., 2018*). Since we find that quotes are only slightly more likely to come from a last author, it is reasonable to compare the relative rate of predicted quotes from men to either authorship position. Comparison with last authorships may reveal more how gender bias currently exists whereas comparison with early career scientists may reveal bias in comparison to a future, more possibly diverse academic environment. We hope that increased representation and recognition of women in science, even beyond what is observed in authorship, can increase the proportion of women first and last authors such that it better reflects the general population.

To further our analysis of possible coverage disparities, we looked at differences in predicted name origins of quoted and cited authors across all the processed news articles. Our use of name origins is a proxy for a journalist's or referring scholarly peer's potential perceptions of the ethnicity of a primary source as signaled by an individual's name. We do not intend to assign an identity to an individual, but to generate a broad metric to measure possible bias for particular ethnicities during journalists' primary source gathering. Our findings provide additional support for previous studies that identified under-citation (*Bertolero et al., 2020*) and under-recognition (*Jan, 2022*) of East Asian people. Interestingly, we found under-citation of people with predicted East Asian name origins to be much less pronounced than under-quotation. We do not believe that the under-quotation is driven by paraphrasing sources, which may occur more frequently with non-native English speakers. We also found that the disparity observed in quotes and mentions was almost eliminated when only considering people who were additionally cited within the same article. This suggests that the source of the disparity may lie in the search for additional expert opinions.

Either way, the clear disparity of predicted East Asian researcher quotes and mentions argues for including a broader set of voices when seeking opinions beyond the academic papers being covered in the article. One solution could be to have region-specific journalists. While we were not directly able to examine the regions journalists lived in, this potential strategy is supported by our analysis of journalists with a predicted East Asian name origin. When considering quotes from people with a predicted East Asian name origin, we found that journalists who themselves have a predicted East Asian name origin include a higher proportion of these quotes than journalists with European or Celtic/English predicted names. When considering only people who were both quoted and cited,

the effect of the predicted name origins of journalists was substantially dampened. We are unable to identify if this is a geographic bias of the reporters in this analysis, since we do not know the location of the journalist at the time of writing the article. As a proxy for measuring possible geographical bias of a journalist, we attempted to identify if there was any geographical bias of cited authors. To do this, we identified the affiliation of each cited author and identified their affiliated country. Unfortunately, we could not robustly extract a large enough number of cited authors from different countries to make any conclusive statements. Expanding our work to other science journalism outlets could help identify possible ways in which geographic region, genders, and perceived ethnicity interact and affect scientific visibility of specific groups. While we are unable to identify that journalists have a specific geographical bias, having reporters explicitly focused on specific regional sources will broaden coverage of international opinions in science.

In our analysis, we also find that there are more first authors with predicted East Asian name origin than last authors. This is in contrast to predicted Celtic/English and European name origins. Furthermore, we see that the amount of first author people with predicted East Asian name origins is increasing at a much faster rate than quotes are increasing. If this mismatched rate of representation continues, this could lead to an increasingly large erasure of early career scientists with East Asian name origins. As noted before, focusing on increasing engagement with early career scientists can help to reduce the growing disparity of public visibility of scientists with East Asian name origins.

Through our comprehensive analysis, we were able to quantify how recognized persons in news journalism vary by name origin and gender, then compare it to scientific publishing background rates. While we found a significant gender disparity compared to the general population, the rate of women's representation in scientific news is increasing and outpacing first and last authorships on scientific papers. Furthermore, we identified a significant reduction of quotes from scientists with a predicted East Asian name origin when compared to paper authorship and a significant but smaller reduction of cited authors with a predicted East Asian name origin in news content.

Computational tools enabled us to automatically analyze thousands of articles to identify existing disparities by gender and name origin, but these tools are not without limitations. Our tools are unable to identify non-binary people and rely on gender predictors that are known to have region-specific biases, with the largest decrease in performance on names of an Asian origin (*Santamaría and Mihaljević, 2018*; *Sebo, 2021*). Furthermore, name origin is only a proxy for externally perceived racial or ethnic origins of a source or author and is not as accurate as self-identified race or ethnicity. Self-identification better captures the lived experience of an individual that computational estimates from a name can not capture. This is highlighted in our inability to distinguish between Black and White people from the United States by their names. As the collection of demographic data by publication outlets grows, we believe this will enable a more fine-grained and accurate analysis of disparities in scientific journalism.

Previous anecdotal studies from journalists have shown that awareness of their bias can help them to reduce it (*LaFrance, 2013*; *LaFrance, 2016*; *Yong, 2018*). Once a bias is identified an individual can seek resources to help them find and retain diverse sources, such as utilizing international expert databases like gage (*Gage, 2024*) and SheSource (*Women's Media Center, 2024*). Additional tips for journalists to achieve and maintain a diverse source pool is described by Christina Selby in the Open Notebook (*Selby, 2016*).

It should also be mentioned that we were only able to analyze the data provided through scraping 'https://www.nature.com/'. This is a major limitation, because the only measures that we have of demographics of sources are people who have their name mentioned or research cited within the article. Journalists do not quote or mention all of the sources that they interviewed or cite all of the papers that they read when researching an article. For example, a person may not be mentioned or quoted in the article because of length limitations, because they do not want to be named, or if they provide information that is not directly quotable but that still shapes the content of the article. A more accurate reflection of journalists' sources would be a self-maintained record of people they interview. Our work examines disparities with respect to recognition within articles, which can be measured by mentions, quotes, or citations of people.

Furthermore, the news articles present on 'https://www.nature.com/' are intended for a very specific readership that may not be reflective of more broad scientific news outlets. In a separate analysis, we took a cursory look into a comparison with *The Guardian* and found similar disparities in

gender and name origin. However, it is not clear which publications should be used as a comparator for science-related articles in *The Guardian*, and difficult to compare relative rates of representation. While other science news outlets may not have a direct comparator, it would be useful to take a broad comparison across multiple science news outlets to compare against one another. Our existing pipeline could be easily applied to other science news outlets and identify if there exists a consistent pattern of disparity regardless of the intended readership.

Another major limitation of our study is that we only used articles published by *Nature* or *Springer Nature* as a comparator. Not all papers are interesting to the general public and likely to be covered by journalists. In this work, we assume that the demographics of scientists publishing work that is likely to be covered by journalists matches the demographics of all scientists publishing articles in *Nature*, *Springer Nature*, or other publishers. Our work could be extended to include additional publishers and pre-print servers. To reveal more scientific-field-specific biases, analyses could be performed on individual topics versus our aggregate analysis.

Furthermore, many journalists are limited by who responds to their requests for an interview or recommendations from prominent scientists. Scientists fielding reporter inquiries can also audit themselves to examine the extent to which there are disparities in the sets of experts they recommend. Journalists and the scientists they interview have a unique opportunity to shape the public and their peers' perspectives on who is a scientific expert. Their choice of coverage topics and interviewees could help to reduce disparities in the outputs of science-related journalism.

## Methods
### Data acquisition and processing
#### Text scraping

We scraped all text and metadata from *Nature* using the web-crawling framework Scrapy (***Scrapy developers, 2020***. Scrapy is a tool that applies user-defined rules to follow hyperlinks on webpages and return the information contained on each webpage. We used Scrapy to extract all web pages containing news articles and extract the text. We created four independent scrapy web spiders to process the news text, news citations, journalist names, and paper metadata. News articles were defined as all articles from 2005 to 2020 that were designated as 'News', 'News Feature', 'Career Feature', 'Technology Feature', and 'Toolbox'. Using the spider 'target_year_crawl.py', we scraped the title and main text from all news articles. We character normalized the main text by mapping visually identical Unicode codepoints to a single Unicode codepoint and stripping many invalid Unicode characters. Using an additional spider defined in 'doi_crawl.py', we scaped all citations within news articles. For simplicity, we only considered citations with a DOI included in either text or a hyperlink in this spider. Other possible forms of citations, e.g. titles, were not included. The DOIs were then queried using the *Springer Nature* API. The spider 'article_author_crawl.py' scraped all articles designated 'Article' or 'Letters' from 2005 to 2020. We only scraped author names, author positions, and associated affiliations from research articles, which we refer to as *papers*. It should be noted that 'News' article designations changed over time and partially explain the changing frequency of news articles within each category. The frequency of 'News' articles decreased, but more specific news-related article types increased. Additionally, scraping for journalist names was performed months after the initial scraping of the text, and some aspects of the *Nature* website changed. Website changes caused us to lose unique file mappings between the scraped journalist name and other article metadata for 137 articles. Less than 30 articles per year were impacted.

#### coreNLP

After the news articles were scraped and processed, the text was processed using the coreNLP pipeline (***Manning et al., 2014***; ***Manning et al., 2020***). The main purpose for using coreNLP was to identify named entities related to mentioned and quoted speakers. We used the standard set of annotators: tokenize, ssplit, pos, lemma, ner, parse, coref, and additionally the quote annotator. Each of which, respectively, performs text tokenization, sentence splitting, part of speech recognition, lemmatization, named entity recognition, division of sentences into constituent phrases, co-reference resolution, and identification of quoted entities. We used the 'statistical' algorithm to perform co-reference resolution for speed. Each of these aspects is required in order to identify the names of quoted

or mentioned speakers and identify any of their associated pronouns. All results were output to json format for further downstream processing.

### Springer Nature API

Springer Nature was chosen over other publishers and search engines for multiple reasons: (1) ease of use; (2) it is a large publisher, second only to Elsevier; (3) it covers diverse subjects, in contrast to PubMed, which focuses on the biomedical and life sciences literature; (4) its API has a large daily query limit (5000/day); and (5) it provided more author affiliation information than found in Elsevier. We generated a comparative background set for supplemental analysis with the *Springer Nature* API by obtaining author information for papers cited in news articles. We selected a subset of papers to generate the *Springer Nature* background set. These papers were the first 200 English-language 'Journal' papers returned by the *Springer Nature* API for each month, resulting in 2400 papers per year from 2005 through 2020. These papers are the first 200 papers published each month by a *Springer Nature* journal, which may not be completely random, but we believe to be a reasonably representative sample. Furthermore, the *Springer Nature* articles are only being used as an additional comparator to the complete set of all *Nature* research papers used in our analyses. To obtain the author information for the cited papers, we queried the *Springer Nature* API using the scraped DOI. For both API query types, the author names, positions, and affiliations for each publication were stored and are available in 'all_author_country.tsv' and 'all_author_fullname.tsv'.

## Name formatting

### Name formatting for gender prediction in quotes or mentions

We first pre-filter articles that have more than 25 quotes, which results in the removal of 2.69% (433/16,080) of the total articles. This was done to ensure no single article is over-represented and to avoid spuriously identified quotes due to unusual article formatting. To identify the gender of a quoted or mentioned person, we first attempt to identify the person's full name. Even though genderizeR, the computational method used to predict the name's gender, only uses the first name to make the gender prediction, identifying the full name gives us greater confidence that we correctly identified the first name. To identify the full name, we take the predicted speaker by coreNLP. Unfortunately, this is not always the full name and is only either the first or last name, with the full name occurring somewhere else in the article. In order to get the full name for all names that coreNLP is unable to identify, we match the coreNLP-identified partial name to the longest matching name within the same article. We match names by finding the longest mentioned name in the article with minimal edit (Levenshtein) distance. The name with the smallest edit distance, where character deletions have zero cost, is defined as the matching name. Character deletion was assigned a zero cost because we would like exact substring matches. For example, the calculated cost, including a cost for character deletion, between John and John Steinberg is 10; without character deletion, it is 0. Compared with the distance between John and Jane Doe, with character deletion cost, it is 7; without it is 2. If we are still unable to find a full name, or if coreNLP cannot identify a speaker at all, we also determine whether or not coreNLP linked a gendered pronoun to the quote. If so, we predict that the gender of the speaker is the gender of the pronoun. We ignore all quotes with no name or partial names and no associated pronouns. A summary of processed gender predictions of quotes at each point of processing is provided in *Table 1*.

### Name formatting for gender prediction of authors

Because we separate first and last authors, we only considered papers with more than one author. Roughly 7% of all papers were estimated to be single authors and removed from this analysis: 1113/15,013 for cited Springer articles, 2899/42,155 for random Springer articles, and 955/12,459 for Nature research articles. As for quotes, we needed to extract the first name of the authors. We cast names to lowercase and processed them using the R package humaniformat (*Oliver Keyes, 2016*). humaniformat is a rule-based program that uses character markers to identify if names are reversed (Lastname, Firstname), find middle names and titles. This processing was not required for quote prediction because names written in news articles did not appear to be reversed or abbreviated. Since many last or first authorships may be non-names, we additionally filtered out any identified names if they partially or fully match any of the following terms: 'consortium', 'group', 'initiative', 'team', 'collab',

'committee', 'center', 'program', 'author', or 'institute'. Furthermore, since many papers only contain first name initials (e.g. 'N. Davidson'), we remove any names less than four letters (length includes punctuation) and containing a '.' or '-', then strip out all periods from the first name. This ensures that hyphenated names are not changed, e.g. Julia-Louise remains unchanged, but removes hyphenated initials, e.g. J-L. A summary of processed author gender predictions at each point of processing is provided in *Tables 2–4*.

## Name formatting for name origin prediction

In contrast to the gender prediction, we require the entire name in all steps of name origin prediction. For names identified in the *Nature* news articles, we use the same process as described for the gender prediction; we again try to identify the full name. For author names, we process the names as previously described for the gender prediction of authors. For all names, we only consider them in our analyses if they consist of two distinct parts separated by a space and excluding titles (e.g. Mrs., Prof., Dr., etc.). All names that were filtered out in the analysis of quotes and mentions are provided on our github in the file 'data/author_data/all_mentioned_fullname_excluded.tsv' and 'data/author_data/all_speaker_fullname_excluded.tsv'. A summary of processed name origin predictions of quotes and citations at each point of processing is provided in *Tables 1–4*.

## Gender analysis

The quote extraction and attribution annotator from the coreNLP pipeline was employed to identify quotes and their associated speakers in the article text. In some cases, coreNLP could not identify an associated speaker's name but instead assigned a gendered pronoun. In these instances, we used the gender of the pronoun for the analysis. The R package genderizeR (*Wais, 2016*), a wrapper for the genderize.io API (*Demografix ApS, 2024*), predicted the gender of authors and speakers. We predicted a name as indicating a man if the first name was predicted by genderizeR to come from a man with at least a probability of 50%. To reduce the number of queries made to genderize.io, a previously cached gender prediction from *Le et al., 2020* was also used and can be found in the file 'genderize.tsv'. All first name predictions from this analysis are in the file 'genderize_update.tsv'. To estimate the gender gap for the quote gender analyses, we used the proportion of total quotes, not quoted speakers. We used the proportion of quotes to measure speaker participation instead of only the diversity of speakers. The specific formulas for a single year are shown in *Equations 1 and 2*. We did not consider any names where no prediction could be made or quotes where neither speaker nor gendered pronoun was associated.

$$\text{Prop. Quotes from Men} = \frac{\left|\text{Speaker Quotes from Men}\right|}{\left|\text{Speaker Quotes from Men or Women}\right|} \tag{1}$$

$$\text{Prop. First Author Men} = \frac{\left|\text{First Authors Men}\right|}{\left|\text{First Author Men or Women}\right|} \tag{2}$$

## Name origin analysis

We used the same quoted speakers as described in the previous section for the name origin analysis. In addition, we also consider all authors cited in a *Nature* news article. In contrast to the gender prediction, we need to use the full name to predict name origin. We submitted all extracted full names to Wiki-2019LSTM (*Le et al., 2020*) to predict one of 10 possible name origins: African, Celtic/English, East Asian, European, Greek, Hispanic, Hebrew, Arabic/Turkish/Persian, Nordic, and South Asian. While a full description of Wiki-2019LSTM is outside the scope of this paper, we describe it here breifly. Wiki-2019LSTM is trained on name and nationality pairs, using 3-mers of the characters in a name to predict a nationality. To ensure robust predictions, nationalities were grouped together as described in NamePrism (*Ye et al., 2017*). NamePrism chose to exclude the United States, Australia, and Canada from their country groupings and were therefore excluded during training of Wiki-2019LSTM. This choice was justified by NamePrism in stating that these countries had a high level of immigration. The treemap of country groupings defined in the NamePrism manuscript is found in figure 5 of the publication (*Ye et al., 2017*).

After running the pre-trained Wiki-2019LSTM model, we used the probability origin for each name instead of a hard assignment to a single class. Hard assignment was not used because it has been shown to reproduce biases due to the underreporting of Black and overprediction of White individuals (*Kozlowski et al., 2022*). Similar to the gender analyses, quote proportions were again directly compared against publication rates, except using the probability of assignment instead of the count of hard assignments. For citations, quotes, and mentions, we calculated the proportion for a given year for each name origin. This is shown in *Equation 3* to, for example, calculate the citation rate for last authors with a Greek name origin for a single year.

$$\text{Prop. Greek Last Author Cited} = \frac{\Sigma(\text{Probability Greek Name for each Cited Last Author})}{\left|\text{Cited Last Authors w/any Name}\right|} \quad (3)$$

$$\text{Prop. Greek Quotes} = \frac{\Sigma(\text{Probability Greek Name for each Quoted Speaker})}{\left|\text{Quotes w/any Named Speaker}\right|} \quad (4)$$

$$\text{Prop. Greek Names Mentioned} = \frac{\Sigma(\text{Probability Greek Mentioned Name})}{\left|\text{Unique Names w/any Origin Mentioned}\right|} \quad (5)$$

## Bootstrap estimations

We used the boostrap method to construct confidence intervals for each of our calculated statistics. For all analyses related to *Equations 1–5*, we independently selected 1000 bootstrap samples for each year. We sampled with replacement of size equal to the cardinality of the complete set of interest. Bootstrap estimates for *Equations 1–5* were performed by sampling the denominator set. The mean, 5th, 95th quantiles across the estimates are reported as the estimated mean, lower, and upper bounds.

## Code availability

This manuscript was written using Manubot (*Himmelstein et al., 2019*) and is available on github: manuscript repository link (copy archived at *Greene Lab, 2024a*). All code and metadata are also available on github: full analysis repository link (copy archived at *Greene Lab, 2024b*) under a BSD 3-Clause License. The code to generate all main and supplemental figures is available as R markdown documents within our main analysis github, in the following subfolder: notebooks. We provide a docker image that can re-run the analysis pipeline using intermediate, pre-processed data and produce all the main and supplemental figures. To re-run the entire pipeline (including scraping), the docker image contains all necessary packages and code. Scraping code is available on our main analysis github, in the following subfolder: scraper. The shell scripts to re-run the entire analysis are provided in the README file in the github repository.

## Acknowledgements

We would like to thank Jeffrey Perkel for asking thoughtful questions that spurred this line of research, and providing feedback and insight into the news-gathering process during the course of this project.

## Additional information

### Funding

| Funder | Grant reference number | Author |
| --- | --- | --- |
| Gordon and Betty Moore Foundation | GBMF 4552 | Casey S Greene |
| National Human Genome Research Institute | K99HG012945 | Natalie R Davidson |

The funders had no role in study design, data collection, and interpretation, or the decision to submit the work for publication.

## Author contributions
Natalie R Davidson, Data curation, Software, Formal analysis, Investigation, Visualization, Methodology, Writing - original draft, Writing - review and editing; Casey S Greene, Conceptualization, Resources, Supervision, Funding acquisition, Project administration, Writing - review and editing

## Author ORCIDs
Natalie R Davidson http://orcid.org/0000-0002-1745-8072
Casey S Greene http://orcid.org/0000-0001-8713-9213

Reviewer #1 (Public Review): https://doi.org/10.7554/eLife.84855.3.sa1
Reviewer #2 (Public Review): https://doi.org/10.7554/eLife.84855.3.sa2
Author response https://doi.org/10.7554/eLife.84855.3.sa3

# Additional files

## Supplementary files
• MDAR checklist

## Data availability
Due to copyright, we are unable to provide the unprocessed scraped data used in this analysis. To ensure reproducibility without violating copyright, we provide the word frequencies for each news article, the coreNLP output, all analyzed names with an article identifier, as well as any other associated data used in the analyses such as quotes and citations. We provide all of this data on github. We also provide data descriptions in our github README, under the header 'Quick data folder overview'. Our data is also uploaded to Zenodo.

The following dataset was generated:

| Author(s) | Year | Dataset title | Dataset URL | Database and Identifier |
| --- | --- | --- | --- | --- |
| Davidson NR, Greene CS | 2022 | Nature News Disparities | https://zenodo.org/record/7387906 | Zenodo, 10.5281/zenodo.7387906 |

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
