## [Editor Report · eLife assessment]

This **important** bibliometric analysis shows that authors of scientific papers whose names suggest they are female or East Asian get quoted less often in news stories about their work. While caveats are inevitable in this type of study, the evidence for the authors' claims is **convincing**, with a rigorous, and importantly, reproducible analysis of over 20,000 articles from across 15 years. This paper will be of interest to science journalists and to researchers who study science communication.

---

## [Referee Report · Reviewer #1 (Public Review)]

I thank the authors for addressing almost all my comments on the previous version of this manuscript, which studies the representation by gender and name origin of authors from Nature and Springer Nature articles in Nature News.

The representation of author identities is an important step towards equality in science, and the authors found that women are underrepresented in news quotes and mentions with respect to the proportion of women authors.

The research is rigorously conducted. It presents relevant questions and compelling answers. The documentation of the data and methods is thoroughly done, and the authors provide the code and data for reproduction.

---

## [Referee Report · Reviewer #2 (Public Review)]

The authors have done well to address the points raised in my previous review.

The updated version of this manuscript retains the technical competence of the first, but with important changes that make the analysis more legible and results better contextualized. Specifically, the discussion is richer, the interpretation of the results is more nuanced, the terminology is more precise, and issues of clarity related to the methodology and results have been resolved.

Broad caveats remain about the nature of authorship, and who we should expect to be quoted in science journalism. Namely, who is the lead author? Ideally, the corresponding author would be included as well, or else some bibliometric definition of the most senior author on the byline. However, the authors' approach here is certainly adequate, and they did well to incorporate discussion of authorship and the scholarly division of labour in their discussion.

In sum, I find the article greatly improved and a competent analysis into the unequal use of quotations in scientific journalism.

---

## [Author Response]

[The following is the authors’ response to the current reviews.]

In response to Reviewer #2, we agree with the reviewer that it needs to be noted that not all forms of recognition are the same and have added the following: "However, we note that not all forms of recognition are the same; researchers may prefer to have their work featured instead of personal stories or critiques of the scientific environment."

[The following is the authors’ response to the previous reviews.]

We thank both reviewers for their detailed comments and insightful suggestions. Below we summarize our responses to each concern in addition to the edits within the manuscript.

We would also like to add a clarification to the eLife assessment, it states “This important bibliometric analysis shows that authors of scientific papers whose names suggest they are female or East Asian get quoted less often in news stories about their work.” We show that individuals with names predicted to be from women or East Asian name origins are less likely to be quoted or mentioned in Nature’s scientific news stories than expected by publication demographics. In this study, we did not compare the level of coverage of a scientific article by the demographics of the authors of the article.

**Reviewer #1**
The article is not so clearly structured, which makes it hard to follow. A better framing, contextualization, and conceptualization of their analysis would help the readers to better understand the results. There are some unclear definitions and wrong wording of key concepts.

We have adapted our wording in the text and added a more detailed discussion which hopefully makes the paper easier to comprehend. These changes are described in the context of your reviewer's suggestions and addressed in the next section.

Language use: Male/Female refers to sex, not to gender.

We have now updated the language throughout the text. Thank you for pointing this out.

Regional disparities are not the same as names' origin. While the first might relate to the academic origin of authors, inferred from their institutional belonging, the latter reflects the authors' inferred identity. Ethnic identities and the construction of prejudice against specific populations need proper contextualization.

We have added better contextualization in the manuscript and reworded the section in our results and discussion to clarify that we are analyzing disparities related to perceived ethnicity and not regions. We also added the following text to the results section “In our analysis, we use name origin as an estimate for the perceived ethnicity of a primary source by a journalist. Our prediction is not intended to assign ethnicity to an individual, but to be used broadly as a tool to quantify representational differences in a journalist's sociologically constructed perception of a primary source's ethnicity.” We also added the following text to our Discussion: “Our use of name origins is a proxy for a journalist's or referring scholarly peer’s potential perceptions of the ethnicity of a primary source as signaled by an individual's name. We do not intend to assign an identity to an individual, but to generate a broad metric to measure possible bias for particular ethnicities during journalists' primary source gathering.”

It would be helpful to have a clear definition of what are quotes, mentions, and citations. For me, it was not so clear and made understanding the results more difficult.

We added the following text to the results section Extracted Data Used for Analysis: “Quoted names are any names that were attached to a quote within the article. Mentioned names are any names that were stated within the article. Cited names are all author names of a scientific paper that was cited in the news article.”

The comparison against Nature published research articles is not perfect because journalists will also cover articles not published in Nature. If for example, the gender representation in the quoted articles is not the same between Nature journals and other journals, then this source of inequality would be missing (e.g. if the journalists are biased against women, but not as much when they published in Nature, because they are also biased towards Nature articles). Also, the gender representation among Nature authors could not be the same as in general. Nevertheless, this seems to be a fair benchmark, especially if the authors did not have access to other more comprehensive databases. But a statement of limitations including these potential issues would be good to have.

To add better context to the generalizability of our work, we added the following text to our discussion: “Furthermore, the news articles present on "https://www.nature.com/" are intended for a very specific readership that may not be reflective of more broad scientific news outlets. In a separate analysis, we took a cursory look into a comparison with The Guardian and found similar disparities in gender and name origin. However, it is not clear which publications should be used as a comparator for science-related articles in The Guardian, and difficult to compare relative rates of representation. While other science news outlets may not have a direct comparator, it would be useful to take a broad comparison across multiple science news outlets to compare against one another. Our existing pipeline could be easily applied to other science news outlets and identify if there exists a consistent pattern of disparity regardless of the intended readership.”

"we select the highest probability origin for each name as the resultant assignment". Threshold based approaches for race/ethnicity name-based inference have been criticized by the literature as they might reproduce biases (see Kozlowski, D., Murray, D. S., Bell, A., Hulsey, W., Larivière, V., Monroe-White, T., & Sugimoto, C. R. (2022). Avoiding bias when inferring race using name-based approaches. Plos one, 17(3), e0264270.). The authors could use the full distribution of probabilities over names instead of selecting one. The formulae proposed (3-5) could be easily adapted to this change.

We thank the author for pointing this out. We have updated our analysis to use the probabilities instead of hard assignments. Figure 3 and formulae 3-5 have been updated. While we observe a slight shift in the calculated values, the overall trends are unchanged.

Is it possible to make an analysis that intersects both name origin and gender? I am not sure if the sample size would allow for this, but if some other dimensions were collapsed, it would be very important to show what happens at the intersection of these two dimensions of discrimination.

We agree that identifying any differences in quotation patterns at the intersection of gender and name origin would be very useful to identify. To address this, we added supplemental table 5. This table identifies the number of quotes per predicted name origin and gender over all years and article types. In this table, we don’t see a significant difference in gender distribution across predicted name origins.

Given a larger sample size, we would be able to better identify more subtle differences, but at this sample size, we cannot make more detailed inferences. Additionally, this also addresses a QC-issue, where predicted gender accuracy varies by name origin, specifically East Asian name origin. From our data, we don’t see a large difference in proportions across any name origin. We added the following text to the results section to incorporate this analysis:

“However, it should be noted that the error rate varies by name origin with the largest decrease in performance on names with an Asian origin [@doi:10.7717/peerj-cs.156;@doi:10.5195/jmla.2021.1252]. In our analysis, we did not observe a large difference in names predicted to come from a man or woman between predicted East Asian and other name origins (Table 5).”

The use of vocabulary should be more homogeneous. For example, in page 13 the authors start to use the concepts of over/under enrichment, which appeared before in a title but was not used.

The text has been updated to remove all mentions of “over/under enrichment” with “over/under representation”

In the discussions section, it would be important to see as a statement of limitations the problems that automatic origin and gender inference have.

We thank the reviewer for this suggestion. We have added the following paragraph to our discussion.

Computational tools enabled us to automatically analyze thousands of articles to identify existing disparities by gender and name origin, but these tools are not without limitations. Our tools are unable to identify non-binary people and rely on gender predictors that are known to have region-specific biases, with the largest decrease in performance on names of an Asian origin[@doi:10.7717/peerj-cs.156;@doi:10.5195/jmla.2021.1252]. Furthermore, name origin is only a proxy for externally perceived racial or ethnic origins of a source or author and is not as accurate as self-identified race or ethnicity. Self-identification better captures the lived experience of an individual that computational estimates from a name can not capture. This is highlighted in our inability to distinguish between Black and White people from the US by their names. As the collection of demographic data by publication outlets grows, we believe this will enable a more fine-grained and accurate analysis of disparities in scientific journalism.

Figures 2a and 3a show that the affiliations of authors and their countries was going to be used in this analysis. Yet, this section is not present in the article. I would encourage the authors to add this to the analysis as it would show important patterns, and to intersect the dimensions of gender, name origin and country.

We were interested in using this analysis in our work, but unfortunately the sample size of cited works in each country was too small to make inferences. If this work was extended to larger scientific outlets to include larger corpora such as The Guardian or New York Times, we think one could be able to make more robust inferences. Since our work only focuses on Nature, we decided not to include this analysis. However, we do include a section in our discussion for future work.

“As a proxy for measuring possible geographical bias of a journalist, we attempted to identify if there was any geographical bias of cited authors. To do this, we identified the affiliation of each cited author and identified their affiliated country. Unfortunately, we could not robustly extract a large enough number of cited authors from different countries to make any conclusive statements. Expanding our work to other science journalism outlets could help identify possible ways in which geographic region, genders, and perceived ethnicity interact and affect scientific visibility of specific groups. While we are unable to identify that journalists have a specific geographical bias, having reporters explicitly focused on specific regional sources will broaden coverage of international opinions in science.”

It is not clear at that point what column dependence means.

The abstract has been updated to state, “Gender disparity in Nature quotes was dependent on the article type.”

**Reviewer #2**

We thank the reviewer for their very detailed and insightful suggestions regarding our analysis and the key caveats that needed better contextualization in our analysis. We went through each major point the reviewer brought up below and included any additional text that was needed.

In some cases, the manuscript lacks consistency in terminology, and uses word choice that is strange (e.g., "enrichment" and "depletion" when discussion representation).

We thank the review for pointing this out, we have removed all instances of depletion/enrichment for over/under-representation

Caveats to Claim 1. So while Claim 1 holds, it does not hold for all comparator sets and for all years. I don't think this is critical of the paper-the authors do discuss the trend in Claim 2-but interpretation of this claim should take care of these caveats, and readers should consider the important differences in first and last authorship.

We thank the reviewer for their detailed feedback on this section. We have added the missing contextualization of our results. In the results section, I changed the figure caption to: “Speakers predicted to be men are sometimes overrepresented in quotes, but this depends on the year and article type.” Added the following paragraph “When considering the relative proportion of authors and speakers predicted to be men, we only find a slight over-representation of men. This overrepresentation is dependent on the authorship position and the year. Before 2010, quotes predicted as from men are overrepresented in comparison to both first and last authors, but between 2010 and 2017 quotes predicted from men are only overrepresented in comparison for first authors. In 2020, we find a slight over-representation of quotes predicted to be from women relative to first and last authors, but still severely under-represented when considering the general population. The choice of comparison between first and last authors can reveal different aspects of the current state of academia. While this does not hold in all scientific fields, first authors are typically early career scientists and last authors are more senior scientists. It has also been shown that early career scientists tend to be more diverse than senior scientists [@doi:10.7554/eLife.60829; @doi:10.1096/fj.201800639]. Since we find that quotes are only slightly more likely to come from a last author, it is reasonable to compare the relative rate of predicted quotes from men to either authorship position. Comparison with last authorships may reveal more how gender bias currently exists whereas comparison with early career scientists may reveal bias in comparison to a future, more possibly diverse academic environment. We hope that increased representation and recognition of women in science, even beyond what is observed in authorship, can increase the proportion of women first and last authors such that it better reflects the general population.”

Generalizability to other contexts of science journalism:

We thank the reviewer for their feedback on the generalizability of our work. We have now added the following text to our discussion to provide the reader with a better context of our results:“To articles presented on "https://www.nature.com/" are intended for a very specific readership that may not be reflective of more broad scientific news outlets. In a separate analysis, we took a cursory look into a comparison with The Guardian and found very similar disparities in gender and name origin. However, it is not clear which publications should be used as a comparator for science-related articles in The

Guardian, and difficult to compare relative rates of representation. While other science news outlets may not have a direct comparator, it would be useful to take a broad comparison across multiple science news outlets to compare against one another. Our existing pipeline could be easily applied to other science news outlets and identify if there exists a consistent pattern of disparity regardless of the intended readership. ”

Shallow discussion:The authors highlight gender parity in career features, but why exactly is there gender parity in this format?

We thank the reviewer for encouraging us to better contextualize our findings in the broader discourse. We have now added several sections to our Discussion. To address gender parity, we have added the following text: “This finding, coupled with the near equal number of articles written by journalists predicted to be men or women, argues for more diversity in topical coverage. "Career Feature" articles highlight current topics relevant to working scientists and frequently highlight systemic issues with the scientific environment. This column allows space for marginalized people to critique the current state of affairs in science or share their personal stories. This type of content encourages the journalist to seek out a diverse set of primary sources. Including more content that is not primarily focused on recent publications, but all topics surrounding the practice of science, can serve as an additional tool to rapidly achieve gender parity in journalistic recognition.”

Representation in quotations varies by first and last author, most certainly as a result of the academic division of labor in the life sciences. However, what does it say about the scientific quotation that it appears first authors are more often to be quoted? Does this mean that the division of labor is changing such that the first authors are the lead scientists? Or does it imply that senior authors are being skipped over, or giving away their chance to comment on a study to the first author?

We thank the reviewer for asking bringing up these important questions. We have added better context to our first author analysis in our discussion. We have included the following two sections to address this. Also, we want to state that we find last authors to be slightly more quoted than first authors, as depicted in Fig. 2d., with first author quotation percentage largely appearing below the red line. We included this text in a response above and include it again here for convenience.

“Before 2010, quotes predicted as from men are overrepresented in comparison to both first and last authors, but between 2010 and 2017 quotes predicted from men are only overrepresented in comparison for first authors. In 2020, we find a slight over-representation of quotes predicted to be from women relative to first and last authors, but still severely under-represented when considering the general population. The choice of comparison between first and last authors can reveal different aspects of the current state of academia. While this does not hold in all scientific fields, first authors are typically early career scientists and last authors are more senior scientists. It has also been shown that early career scientists tend to be more diverse than senior scientists [@doi:10.7554/eLife.60829; @doi:10.1096/fj.201800639]. Since we find that quotes are only slightly more likely to come from a last author, it is reasonable to compare the relative rate of predicted quotes from men to either authorship position. Comparison with last authorships may reveal more how gender bias currently exists whereas comparison with early career scientists may reveal bias in comparison to a future, more possibly diverse academic environment. We hope that increased representation and recognition of women in science, even beyond what is observed in authorship, can increase the proportion of women first and last authors such that it better reflects the general population.”

“In our analysis, we also find that there are more first authors with predicted East Asian name origin than last authors. This is in contrast to predicted Celtic/English and European name origins. Furthermore, we see that the amount of first author people with predicted East Asian name origins is increasing at a much faster rate than quotes are increasing. If this mismatched rate of representation continues, this could lead to an increasingly large erasure of early career scientists with East Asian name origins. As noted before, focusing on increasing engagement with early career scientists can help to reduce the growing disparity of public visibility of scientists with East Asian name origins.”

What might be the downstream impacts on the public stemming from the under-representation of scientists with East Asian names? According to Figure 3d, not only are East Asian namesunder-represented in quotations, but they are becoming more under-represented over time as they appear as authors in a greater number of Nature publications; Those with European names are proportionately represented in quotations given their share of authors in Nature. Why might this be, especially seeing as Anglo names are heavily over-represented?

To address this point, we have added the following text to our discussion: “In our analysis, we also find that there are more first authors with predicted East Asian name origin than last authors. This is in contrast to predicted Celtic/English and European name origins. Furthermore, the amount of first author people with predicted East Asian name origins is increasing at a much faster rate than quotes are increasing. If this mismatched rate of representation continues, this could lead to an increasingly large erasure of early career scientists with East Asian name origins. As noted before, focusing on increasing engagement with early career scientists can help to reduce the growing disparity of public visibility of scientists with East Asian name origins.”

I am very confused by Figure 1B. It mixes the counts of News-related items with (non-Springer) research articles in a single stacked bar plot which makes determining the quantity of either difficult. I would advise splitting them out.

Figure 1B has been updated, and the News and Research articles have been separated.

When querying the first 2000 or so results from the SpringerNature API, are the authors certain that they are getting a random sample of papers?

These papers were the first 200 English language "Journal" papers returned by the *Springer Nature* API for each month, resulting in 2400 papers per year from 2005 through 2020. These papers are the first 200 papers published each month by a *Springer Nature* journal, which may not be completely random, but we believe to be a reasonably representative sample. Furthermore, the *Springer Nature* comparator set is being used as an additional comparator to the complete set of all *Nature* research papers used in our analyses.

In all figures: the authors use capital letters to indicate panels in the caption, but lowercase letters in the figure itself and in the main text. This should be made consistent.

This has been updated.

In all figures: the authors should make the caption letter bold in the figure captions, which makes it much easier to find descriptions of specific panels

This has been updated.

In the section "coreNLP": the authors mention "co-reference resolution" but without really remarking why it is being used. This is an issue throughout the methods-the authors describe what method they are using but either they don't mention *why* they are using that method until later, or else not at all.

We have added better reasoning behind our coreNLP selected methods: “We used the standard set of annotaters: tokenize, ssplit, pos, lemma, ner, parse, coref, and additionally the quote annotator. These perform text tokenization, sentence splitting, part of speech recognition, lemmatization, named entity recoginition, division of sentences into constituent phrases, co-reference resolution, and identification of quoted entities, respectively. We used the "statistical" algorithm to perform coreference resolution for speed. Each of these aspects is required to identify the names of quoted or mentioned speakers and identify any of their associated pronouns. All results were output to json format for further downstream processing.”

We included a better description of Scrapy: “Scrapy is a tool that applies user-defined rules to follow hyperlinks on webpages and return the information contained on each webpage. We used Scrapy to extract all web pages containing news articles and extract the text.”

We also included our motivation for bootstrapping: “We used the boostrap method to construct confidence intervals for each of our calculated statistics.”

In the section "Name Formatting for Gender Prediction in Quotes or Mentions", genderizeR is mentioned before an introduction to the tool

We added the following text to provide context: “Even though genderizeR, the computational method used to predict the name's gender, only uses the first name to make the gender prediction, identifying the full name gives us greater confidence that we correctly identified the first name. “

In the section "Name Formatting for Gender Prediction of Authors", you state that you exclude papers with only one author. How many papers is this? I assume few, in Nature, but if not I can imagine gender differences based on who writes first-authored papers.

We find that the number excluded is roughly 7% of all papers, which is consistent across Nature and Springer Nature (1113/15013 for cited springer articles, 2899/42155 for random springer articles, 955/12459 for nature authors). We have added the following text to the manuscript for better context: “Roughly 7% of all papers were estimated to be by a single author and removed from this analysis.: 1113/15013 for cited Springer articles, 2899/42155 for random Springer articles, 955/12459 for Nature research articles.”

In "Name Origin Analysis", for the in-text reference to Equation 3: include the prefix "Eq." or similar to mark this as referencing the equation and not something else.

This has been updated.

The use of the word "enrichment" in reference to the representation of East Asian authors is strange and does not fit the colloquial definition of the term. I suggest just using a simpler term like "representation" instead.Similarly, the authors use the word "depletion" to reflect the lower rate of quotes to scientists with East-Asian names, but I feel a simpler word would be more appropriate.

We thank the reviewer for this suggestion, all instances of “enrichment/depletion” have been replaced with “over/under representation”

The authors claim in Figure 2d that there is a steady increase in the rate of first author citations, however, this graph is not convincing. It appears to show much more noise than anything resembling a steady change.

We have reworded our figure description to state that there is a consistent bias towards quoting last authors. Our figure description now states: “Panel d shows a consistent but slight bias towards quoting the last author of a cited article than the first author over time.”

Supplemental Figures 1b and 1c do not seem to be mentioned in the main text, and I struggle to see their relevance.

We thank the reviewer for identifying this error; these subpanels have been removed.